# Implementation of a New Electronic Liquid Dispensing System for Individualized Compounding of Hard Capsules

**DOI:** 10.3390/pharmaceutics14081580

**Published:** 2022-07-29

**Authors:** Bakul Sarker, Mariele Fligge, Tanja Knaab, Jörg Breitkreutz

**Affiliations:** Institute of Pharmaceutics and Biopharmaceutics, Heinrich Heine University, Universitätsstr. 1, 40225 Duesseldorf, Germany; bakul.sarker@hhu.de (B.S.); mariele.fligge@hhu.de (M.F.); tanja.knaab@hhu.de (T.K.)

**Keywords:** personalized medication, liquid dispensing system, low dose drug, capsules, compounding

## Abstract

An automated compounding device can be a useful tool for the rapid and accurate production of small batches for personalized medicine as well as for clinical batches. A novel electronic liquid dispensing system (ELDS) was investigated to produce hard capsules with individualized dose strengths. An ethanol-based solvent system containing the antihypertensive enalapril maleate was extruded through a cannula into prefilled capsules. The capsules were prefilled with a powder bed of mesoporous silica (Syloid 244 FP) or synthetic dibasic calcium phosphate anhydrous (Fujicalin). The dosing accuracy as well as content uniformity of ELDS was compared with manual preparation using a Hamilton syringe (HS). Both methods met the pharmacopeia criteria for all formulations with an acceptance value (AV) less than 15. Drug adherence to the capsule shells was also investigated. A recovery rate of 98% of enalapril maleate showed almost no drug loss, but the appropriateness of the new dispensing method.

## 1. Introduction

Individual compounding of solid oral dosage forms, such as capsules and suppositories, still plays an important role in community and hospital pharmacies [1,2]. To manually manufacture individual hard gelatin capsules is little effort, thus they are often the formulation of choice for very small batches, not only for individualized medicine but also for preformulation and phase I clinical trials [3]. They are easy to handle and have favorable properties such as taste masking [4], product protection, or the ability to transfer the contents if needed. However, there is a lack of safe, on-demand manufacturing of capsules in small batch sizes [5]. In most cases, such as in community pharmacies, they are made by hand. The drug substance, or a crushed solid marketed drug product, is blended and diluted with different excipients for better dosing and filled into the capsule shells with the help of special filling boards or syringes [6]. During powder preparation and filling of the capsules, various errors can occur, e.g., inadequate mixing or powder filling. With syringes there is a risk of incorrect dosing due to handling errors. The consequences are over or under-dosing, which bears many risks, in particular drugs with a narrow therapeutic window. Due to the lack of availability of appropriate dosage forms for the pediatric population, off-label use is very common, especially for antihypertensives, bringing the need for personalized formulations into greater focus [7]. Another problem in pediatrics is inadequate administration form, since small children are often not yet able to swallow solid dosage forms [8]. When not intended to swallow as a whole, capsules can serve as the primary packaging, offering the ability to introduce the powder into a suitable dispensing medium, such as water or milk. However, if the capsule shell is the primary packaging material, only the content of the capsule can be used to prepare a solution or suspension, which simplifies the application in children.

The objective of this study was to implement a new semi-automatic electronic liquid dispensing system (ELDS) to produce formulations with individual dose strengths by extrusion of liquids into capsule shells. This novel electronically automized equipment [9] was developed and customized by Waldeck & Saar under the continuous advice of our working group. There are already several automatic dosing machines to fill capsules with powders and liquids in laboratory or industrial practice. However, these machines are built for industrial manufacturing, e.g., soft capsules, which require an enormous amount of equipment and intensive supervision by trained personnel. The new low-cost simple device presented in the present study has been designed to produce small batches at hospital or community pharmacies in a short time frame.

The instrumentation of the ELDS is shown in Figure 1. The equipment consists of a control unit (Figure 1C), which is a software programming unit for the machine’s operation. The liquid dosing unit (Figure 1B) consists of a syringe holder suitable for disposable syringes and can move in *x* and *z*-axis directions through the travel rails. The capsule board holder, suitable for common capsule board types, can move forwards and backwards through the guide rail.

Figure 1C shows that during the matrix generation the syringe used and needle profile can be selected. In this study a 1000 µL syringe with needle (single use cannula, blunt) was implemented in this machine. However, as the syringe holder (Figure 1B) is 3D printed, it offers the flexibilities for incorporation of other syringe types as well. The travel path for liquid dosing unit is dependent on how the matrix was created for dosing. It also provides the option for selecting the starting and end positions of capsule dosing. For example, one can create a matrix for dosing into 60 capsules but later, the operator can decide from which number of capsule dosing can be started and until which number it should stop dosing. A calculator is implemented in the software program to calculate the required volume of solution for the selected positions for dosing or number of capsules. There is an option for initial extrusion in the matrix which ensures the filling of the void space in the needle and thus facilitates more precise dosing.

A drug-containing liquid is applied directly to the carrier material inside the capsule. To deliver a safe product, a syringe (1000 µL) is used to ensure uniform and precise extrusion of the required volume. Furthermore, a capsule board, which most pharmacies already own, can be used as capsule shells holder. Within the scope of this research, the machine was qualified following the guidelines of GMP. To evaluate the efficiency of the electronic liquid dispensing system (ELDS), all experiments were also conducted with manual filling of capsules using a Hamilton syringe (HS) as comparison. Hamilton syringes have shown high precision compared to other liquid dosing devices, such as Eppendorf pipettes, for liquid dispensing into capsules [10]. As an example, a formulation suitable for pediatric use should be investigated. The anti-hypertensive enalapril maleate with a single dose of 1 mg was used as model drug substance, which was dissolved in ethanol. Syloid 244 FP and Fujicalin were used as carrier materials for prefilling the hard gelatin capsules. To verify possible migration of the drug substance into the capsule shell, the drug content deposited on the capsule shells was also investigated after preparation.

## 2. Materials and Methods

### 2.1. Materials

#### 2.1.1. Chemicals and Reagents

Enalapril maleate was purchased from Zhejiang Huahai Pharmaceutical Company (Taizhou, China). The chemical reference substance enalaprilat was bought from EDQM. Hard gelatin capsules (Coni Snap), size 0, were donated by Lonza-Capsugel (Bornem, Belgium). Silicon dioxide (Syloid 244 FP) was purchased from Grace GmbH & KG (Worms, Germany). Dibasic calcium phosphate, anhydrous (Fujicalin), was bought from Fuji Chemical (Toyama, Japan). Ethanol was purchased from in house chemical supply. Acetonitrile (HPLC grade) and sodium dihydrogen phosphate dihydrate (for analysis) were obtained from Honeywell Research Chemicals (Schwerte, Germany) and Merck (Darmstadt, Germany), respectively, for the HPLC analysis of capsules. Distilled water was obtained by in-lab distillation of demineralized water. The disposable syringe filter (Chromafil Xtra PA- 45/25) was purchased from Macherey-Nagel (Dueren, Germany).

#### 2.1.2. Devices

The electronic liquid dispensing system (ELDS) was received from Waldeck & Saar (Heusweiler, Germany). Disposable syringes (0.01–1 mL) and disposable needles were purchased from B. Braun Melsungen (Melsungen, Germany). The capsule filling device for size 0 capsules (Aponorm) was purchased from Wepa (Montabaur, Germany). The microliter syringe (100 μL, also known as Hamilton syringe) was purchased from Hamilton Bonaduz (Bonaduz, Switzerland).

### 2.2. Methods

#### 2.2.1. Manufacturing of Capsules with Different Dosing Devices

The enalapril maleate 1-mg capsule was produced using the ELDS with two excipients (Syloid 244 FP, Fujicalin) as capsule filler (see Table 1). To compare the feasibility and proficiency of this equipment in individualized capsule dosing, a Hamilton syringe (HS) was used to dose the API following the same working principle.

#### 2.2.2. Formulation of Enalapril Maleate Capsule

The capsule’s basic formulation was to fill the capsules with an excipient (Syoid 244 FP or Fujicalin) and add a volume of API solution into individual capsules. The API solution was prepared in a volatile solvent (absolute ethanol). In order to obtain the targeted dose (1 mg), a volume of 100 μL API solution was added to each capsule. For this, API solutions for dosing into the capsules were prepared with a concentration (conc.) of 10 mg/mL.

#### 2.2.3. Preparation of API Dosing Solution (Enalapril Maleate Stock Solution 10 mg/mL)

For API dosing in capsules, a stock solution of enalapril maleate in absolute ethanol (10 mg/mL) was prepared. Three different batches of stock solutions were prepared using the same method for dosing into the capsules.

#### 2.2.4. Capsule Production Using ELDS

A matrix has to be created in the software programming unit beforehand with precise positions for each capsule inserted in the capsule board, along with the syringe and needle going to be used while dosing. The capsule board must always be placed in the same position. During the matrix creation, positions for each capsule in the capsule board, travel height and liquid extrusion height of the dosing unit must be set for each matrix individually.

While producing capsules using ELDS, a matrix was selected first. The extrusion volume of liquid (100 μL in this case to achieve 1 mg dose of EM per capsule), along with the syringe’s initial extrusion volume, retraction volume and the number of dosing positions, was selected. The manual capsule filling device containing capsules prefilled with excipient without any active substances is placed on the machine’s capsule board holder. The API solution was drawn up manually using the disposable syringe, placed on the syringe holder and fixed with the screws. After completion of the whole setup, the machine was utilized for extrusion. The dosing unit then starts to move over the capsule board through the guide rails and delivers the dictated amount of API solution into each capsule. The capsules were left open for at least 180 min to ensure almost complete evaporation of the ethanol.

#### 2.2.5. Enalapril Maleate Capsule Production by API Dosing Using Microliter Syringe

For API dosing, 100 μL of enalapril maleate stock solution (conc. 10 mg/mL) was removed in succession using the microliter syringe and the solution was manually injected centrally on the filler bed of the capsules body individually for each capsule. Thus, each capsule was dosed with 1 mg of API. The capsules were left open for at least 180 min to ensure almost complete evaporation of the ethanol. Then the capsules were closed. The finished capsules were stored in quick-release polyethylene bags for further analysis.

#### 2.2.6. Analytical Method for Content Uniformity of Dosage Units of Enalapril Maleate 1 mg Capsules

The capsules produced by using different API dosing systems were tested for content uniformity of dosage units according to Ph. Eur. 10.0, monograph 2.9.40. Ten capsule units from each batch were checked for content determination using a validated HPLC analytical method. However, deviating from the conventional method, only the capsule content without the capsule shells was tested. In addition, to investigate the possible loss of API due to adhesion on the capsule shell, after removing the capsule contents, the API content on the capsule shells was also determined using the same HPLC method.

An Elite LaChrom system was used (Hitachi-VWR, Darmstadt, Germany) for the HPLC analysis. The system was equipped with a L2400 UV detector, an L-2300 column oven, and an L-2200 autosampler. The analysis was performed with a Eurosphere II 100-5 C18A column (150 × 4.6 mm) from Knauer Wissens (Germany) as stationary phase. The mobile phase consists of 60% (V:V) phosphate buffer, pH = 2.2, and 40% (V:V) acetonitrile. The oven temperature was 30 °C and the flow rate was 0.7 mL/min. The injection volume was 10 μL and detection was carried out at a wavelength of 215 nm. The runtime of the method was 10 min. The targeted analyte (enalapril) was eluted at approx. 4.3 min, and maleic acid was co-eluted at 2.3 min.

#### 2.2.7. X-ray Powder Diffraction (XRPD)

The X-ray powder diffraction (XRPD) patterns of pure enalapril maleate, Syloid 244 FP and Fujicalin were recorded using a Rigaku MiniFlex X-ray powder diffractometer (Japan) using a 600 W X-ray tube and a D/teX Ultra silicon strip detector, over a θ range of 2–50°. The capsule content (dried-dosed enalapril maleate filling into Syloid 244 FP or Fujicalin powders) was also investigated to examine the solid state of dosed enalapril maleate.

#### 2.2.8. Particle and Surface Morphology

To characterize the adsorption of API onto the surface of the excipients, polarized light microscopy was used. The photographs of pure API, excipients and the mixture of API-excipients were taken with a Leica polarizing light microscope.

#### 2.2.9. In Vitro Drug Release Study

A preliminary in vitro dissolution study was performed according to the Ph. Eur. 10.0, monograph 2.9.3, using configuration 2 (paddle apparatus). A phosphate buffer solution of pH 6.8 was prepared and the dissolution fluid was maintained at 37 ± 0.5 °C with rotation of 60 rpm. As soon as the medium had equilibrated to a stable temperature of 37 °C, the dissolution process was started by placing one capsule per vessel. Six capsules prepared with each excipient were tested. Five milliliters of samples were taken at time points: 1, 3, 6, 10, 15, 20 and 30 min. The samples were analyzed using the HPLC method (see Section 2.2.6) to determine the enalapril maleate concentration in the medium.

#### 2.2.10. Loss on Drying

Five enalapril maleate capsules made of Syloid 244 FP, and five with Fujicalin, were taken as samples for LOD analysis. Only the capsule contents without the capsule shells were placed in the infrared moisture analyzer from Sartorius (Göttingen, Germany) at 80 °C. The amount of evaporated moisture content in percentage obtained from the equipment was recorded.

## 3. Results and Discussion

### 3.1. Preliminary Tests on Formulation for Enalapril Maleate Capsules

The basic formulation of the capsules was to prefill the capsules with a good ab/adsorbing excipient and to add API solution into the prefilled lower capsule parts. After a drying time of 180 min, the capsules were closed and stored for further analysis. To find a suitable model formulation to test the intended functionality of the ELDS, preformulation studies using different excipients, e.g., mannitol, mixture of mannitol-silica (ratio: 99.5:0.5, 98:2, 95:5, 90:10 m/m), Syloid 244 FP and Fujicalin; different solvents: water (20–100 µL) and ethanol (20–100 µL) and with two different capsule shells, hard gelatin capsule shell (Coni-Snap, size 0) and HPMC capsule shell (Vcaps Plus, size 0), were carried out (see Figure 2).

Syloid 244 FP and Fujicalin were finally chosen as excipients for the capsule formulation. A synthetic amorphous silicon dioxide (Syloid 244 FP) was chosen for its high internal mesoporosity and surface area, high adsorptive capacity, high bulk density (for lower dust) and anti-tacking properties [8]. Fujicalin is a synthetic dibasic calcium phosphate anhydrous (DCPA) with high porosity and extremely high specific surface area owing to better adsorption capacity [9]. The API solution was prepared in a volatile solvent, (absolute ethanol), with a concentration of 10 mg/mL of enalapril maleate (EM). Three batches of API dosing solution were prepared, and each batch of API solution was dosed into both excipient-filled capsules (for example, one EM dosing solution named as ‘Stock solution 1′ (B1) was dosed into 10 capsules prefilled with Syloid 244 FP and 10 capsules were prefilled with Fujicalin).

### 3.2. Comparison of Enalapril Maleate Capsule Production with ELDS and Hamilton Syringe

While evaluating the ELDS, the reproducibility and susceptibility of errors in the production method were investigated. Furthermore, the easiness of different dosing methods was assessed by visual observations and investigating abnormalities during production. According to Ph. Eur. 10.0, monograph 2.9.40, the content uniformity of dosage units was tested up to the first test level where the capsule content without the capsule shell was accepted as a single dosage form and the capsule shell was considered as a primary packaging material. The results of the test for content uniformity of dosage units according to Ph. Eur. 10.0, monograph 2.9.40, are shown below in Figure 3A,B.

The average content in percentage and the acceptance value (AV) for each batch of EM capsules with two different excipients are shown in Figure 3. According to Ph. Eur. 10.0, the content uniformity test requirement was met for the capsules produced with both excipients at the first test level. Although the dosed EM in ethanolic solution is adsorbed by only the upper portion of the powder bed in the capsules (Figure 4), the AV value for API content required by the Ph Eur. 10.0 (≤15) was still achieved in all batches at the first test level. The inhomogeneous distribution of solution sometimes leads to the deposition of EM on the capsule shell, resulting in higher standard deviations in API content (%). The standard deviations (SD) obtained from the ELDS are higher compared to those from the HS. One possible reason for the higher SD could be the liquid extrusion pressure from the equipment’s syringe. When the syringe extruded EM solution faster than the Hamilton syringe, the liquid was well adsorbed by Fujicalin compared to Syloid 244 FP. During Hamilton dosing it was observed that liquid extrusion from the syringe was slow as the pressure on the syringe plunger could be controlled with manual handling, and thus each drop of EM solution could receive more time to be adsorbed by the filler substances. Syloid 244 FP is a material with high bulk density and comparatively less porosity than Fujicalin [11,12]. Therefore, while adding the API solutions on the capsule filler bed, Syloid 244 FP resulted in slower adsorption of liquid due to less porosity compared to Fujicalin and it becomes lumpy within the shell due to wetting, which led to a problem called the “Coffee ring effect” [13]. The term ‘Coffee ring effect’ is a phenomenon where a liquid drop dries on a solid surface, and all the suspended particulates are deposited in a ring-like structure. This led to more dose variations in each batch of capsules made with Syloid 244 FP using the ELDS as only the capsule contents were checked to determine the API content, and the API deposited on the capsule shells was not considered.

While comparing the API content (%) and AV value of the EM capsules, both techniques, the semi-automatic ELDS and the manual dosing by Hamilton syringe delivered uniform dosed capsules. However, the standard deviation (%) is higher in capsules manufactured with the ELDS compared to manual manufacturing with Hamilton syringe. One possible reason for this could be the deposition of EM on the capsule shell. With ELDS, the EM solution was extruded onto the capsule filler bed with a higher speed compared to the Hamilton syringe. The capsules containing filler with higher adsorptive capacity (Fujicalin in this case) were able to take up the EM solution faster and did not allow the API solution to deposit on the capsule shell. Furthermore, only the capsule content without capsule shell was analyzed for the content uniformity test; API deposited on the capsule shell was not considered for content calculation. This results in comparatively higher content variation in capsules dosed with the ELDS than the Hamilton syringe.

### 3.3. Determination of Enalapril Maleate Content Deposited on the Capsule Shell

To investigate the possible loss of EM due to deposition on the capsule shell, after removing the capsule contents, the API contents on the capsule shells were determined. For this, ten capsules were produced with Syloid 244 FP, and API dosing was performed using a Hamilton syringe. Two separate analyses were performed to determine the EM content of the capsules. First, the capsule content was removed, dissolved in water and the resulting suspension was analyzed. The remaining capsule shells were numbered accordingly and dissolved in water. The deposited amount of API was checked using the same method. The results are shown in Figure 5.

The result shows that only a small amount of enalapril maleate was deposited on the capsule shell except for one capsule. The recovery rate of targeted dose is still higher than 98% on average, thus this high deposition of API on one capsule could have less significant impact on dose variation.

### 3.4. Evaluation of Drug Release Study

Enalapril maleate is a white crystalline powder, freely soluble in methanol, soluble in ethanol and sparingly soluble in water [14,15]. Since the API was first dissolved in ethanol (absolute) and dosed onto the capsule fillers and dried afterwards to prepare the finished product, during this drying phase from the soluble state, the solid state of dosed EM, whether it was recrystallized or amorphous, was examined. From the XRPD data (Figure 6) it was clearly observed that the pure enalapril maleate exhibits highly intense, narrow, sharp and distinctive diffraction peaks indicating the stable crystalline structure of pure API. However, when the drug was dosed onto the powder bed (Syloid 244 FP or Fujicalin) of capsules, the capsule content prepared with Syloid 244 FP had no sharp distinctive diffraction peaks in the diffractogram due to the amorphous nature of the excipient and amorphous transformation of crystal EM. Nevertheless, the diffractograms from capsules prepared with Fujicalin and enalapril maleate showed some principle peaks with lower intensity. These peaks were from the crystalline structure of Fujicalin, not from the enalapril maleate. As the dosed enalapril maleate was bound to both fillers in amorphous form instead of a crystalline structure, no delayed dissolution or reduced bioavailability would be expected. This was confirmed with a preliminary dissolution study which showed 100% drug release after one minute. Moreover, the images obtained from the polarized microscopy analysis (Figure 7) also reflect the findings discussed above. For drug substances which might undergo changes of the solid state or are even poorly soluble in gastrointestinal fluids, the introduced methodology has to be carefully considered and validated. In the present study, we want to show the principles of the newly developed electronic device.

### 3.5. Loss on Drying

To check the residual solvent amount in the finished products, mass loss due to drying was checked as ethanol is a solvent of class 3 Ph. Eur. (low toxicity) [16]. It was found that capsules made with Fujicalin had an average loss due to drying of 0.36 ± 0.2%, and capsules made with Syloid 244 FP had a much higher value of 5.578 ± 0.5%. Here, the loss due to drying can be composed of both ethanol and water. Additionally, Syloid 244 FP is a highly porous micronized silica powder which is capable of adsorbing a considerable amount of moisture (even from the hard gelatin capsule shell wall) [8]. Thus, it can be concluded that ethanol concentration should be as low as possible in the capsules made with Syloid 244 and the formulation with Fujicalin would be recommended.

## 4. Conclusions

The implementation of the ELDS was successful with respect to the requirements of uniformity of the content of capsules which is often a general problem for low-dosed capsule formulations manually prepared in hospital and community pharmacies. All capsule preparations met the AV value at the first test level given by the Ph. Eur. 10.0, monograph 2.9.40, and provided accurate dosing of 1 mg enalapril maleate. It was demonstrated that the ELDS is suitable to reliably produce a small amount of individually low-dosed capsules in a short time and with low equipment requirements.

Compared to the Hamilton syringe, the ELDS delivers equally precise results. The great advantage of this device is that the probability of errors is significantly lower due to its semi-automation. The Hamilton syringe must be reloaded per capsule as it only holds 100 µL. In addition, the operator must be skilled with Hamilton syringe dosing to achieve good dosing accuracy. This training is comparatively time-consuming. The ELDS, on the other hand, can dose multiple capsules with one syringe elevator and is not dependent on operator experience. The syringe elevator itself can also be automated in the future. Because the Hamilton syringe is used in a reusable manner, a validated clean-up process is needed to eliminate cross-contamination of drug substances. This can be avoided by using disposable syringes, which saves time and money and increases the safety of the product.

To produce individually dosed capsules by extrusion of liquids, it is important to avoid swelling of the capsule shell and possible migration of the drug into it. Otherwise, the contents of the capsules cannot be quantitatively transferred into the desired medium. It was shown that due to the high extrusion accuracy of the machine, highly concentrated drug solutions can be dosed precisely into the prefilled capsules. This results in a minimal amount of liquid that must be injected onto the carrier material to achieve the desired dose leading to a reduced risk of swelling or destruction of the capsule shell. Since the solvent, in this case ethanol, evaporates within a short time, further instability due to moisture is unlikely. However, as the moisture content in capsules made with Syloid 244 FP did not meet the requirements of the guideline, quantitative analysis of ethanol content in capsules should be rechecked. The risk of drug entrapment in the capsule shell could be reduced or could not be visualized and quantitative transfer of the capsule contents ensured. Additionally, after dosing, the EM was no longer in crystalline form but in amorphous form, thus the release of the adsorbed drug from the excipients was faster and complete drug release was achieved.

The used excipients for prefilling, Syloid 244 FP and Fujicalin behaved inertly towards the applied drug solution and the capsule shell. No drug-excipient interactions were detected. The adsorption properties of the fillers led to low contact of the drug solution with the capsule shells. Therefore, low migration of drug into the capsule shells could be observed. In general, other capsule sizes or completely different substrates that are printed with liquid are also possible. Examples could be plasters, gels, orodispersible films or wound dressings. Since many new drugs belong to BCS classes II or IV, the application of lipophilic drug substances with the help of oily liquids would also be conceivable and will be tested in the future.

With this ELDS, individualized capsules with different dose strengths may be produced in a repeatable and reproducible manner within a short time period. This procedure enables rapid individual dosing of capsules with minimal errors and, thus, fulfils the Ph. Eur. requirements for uniformity of dosing units. In general, this production method for individualized capsule preparations is much faster than the conventional methods used in pharmacies. It remains to be investigated whether the presented method is applicable to all drug substances, e.g., those that undergo polymorphic changes or show poor solubility in gastrointestinal fluids. Solubility in the dosing liquid also plays an important role. Dosing of suspension dosing could be attempted to overcome such challenges. Here, special attention has to be paid to the stability of the suspension. Furthermore, other carrier materials should be tested for their suitability. Limitations have to be tested regarding the key properties of the dosing liquid, e.g., dynamic viscosity and surface tension. Currently, the number of capsules to be produced is limited by the volume of the syringe for dosing. The use of larger syringes should be investigated in order to potentially increase the batch size. The syringe holder of the ELDS can be easily adapted, e.g., by three-dimensional printing, but the total mass of the moving syringe could be a limitation for the operating system.

## Figures and Tables

**Figure 1 pharmaceutics-14-01580-f001:**
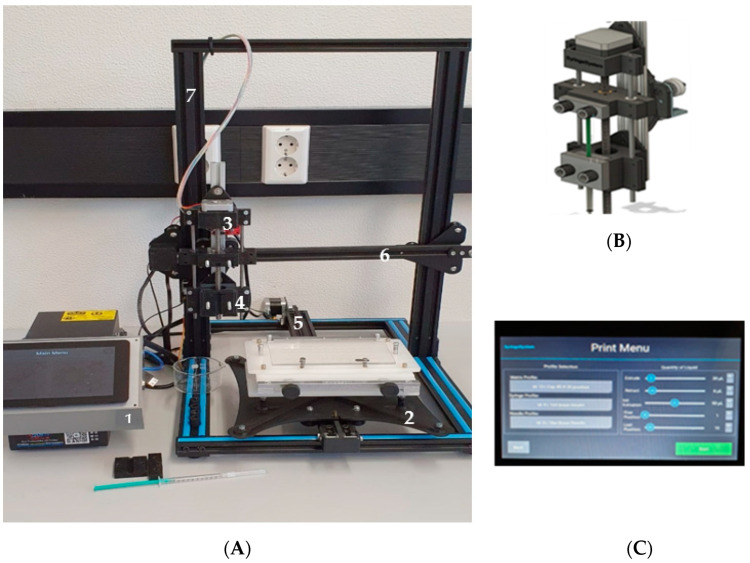
(**A**) Instrumentation of electronic liquid dispensing system (ELDS) consists of (1) control unit, (2) capsule board holder, (3) liquid dosing unit, (4) syringe holder, (5) capsule board travel rail, (6) dosing unit travel rail for *x* and *y*-axis, (7) dosing unit travel rail for *z*-axis; (**B**) liquid dosing unit; (**C**) control unit.

**Figure 2 pharmaceutics-14-01580-f002:**
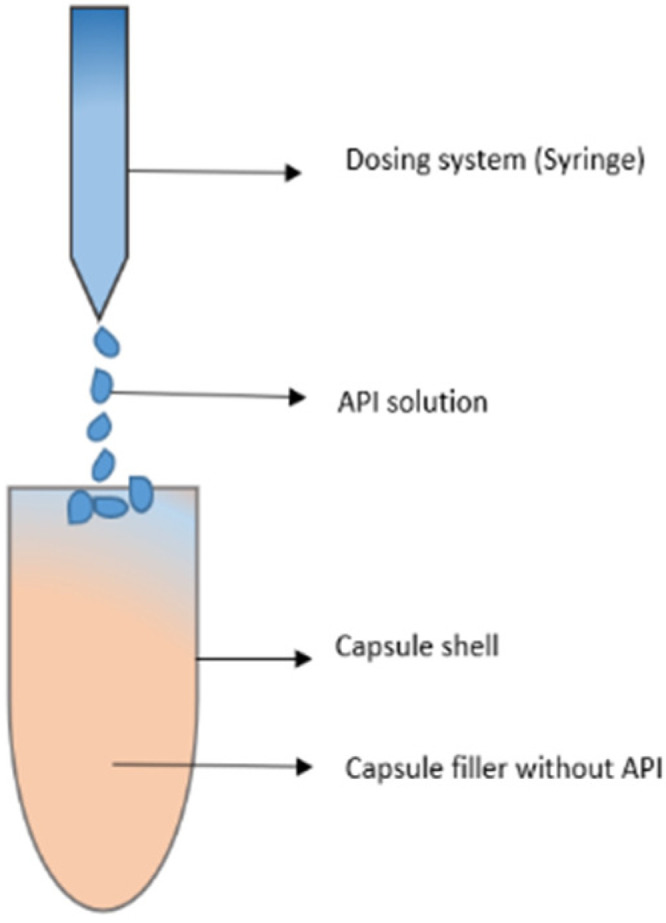
Schematic presentation of API dosing methodology.

**Figure 3 pharmaceutics-14-01580-f003:**
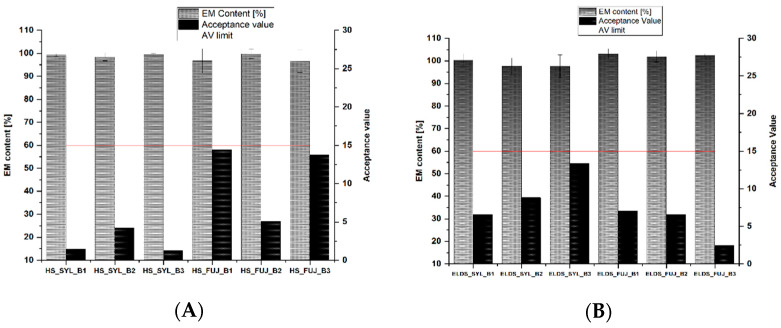
EM content (%) and acceptance value (AV) for 1 mg enalapril maleate capsules. (**A**) Manual capsule dosing using a Hamilton syringe, (**B**) capsule production with ELDS, $\bar{x}$ ± sd, *n* = 10. (*x*-axis labelling: dosing system excipient batch, HS = Hamilton syringe, SYL = Syloid 244 FP, FUJ = Fujicalin, B = batch), red line indicates the specified acceptance value limit for level 1 testing according to Ph.Eur.

**Figure 4 pharmaceutics-14-01580-f004:**
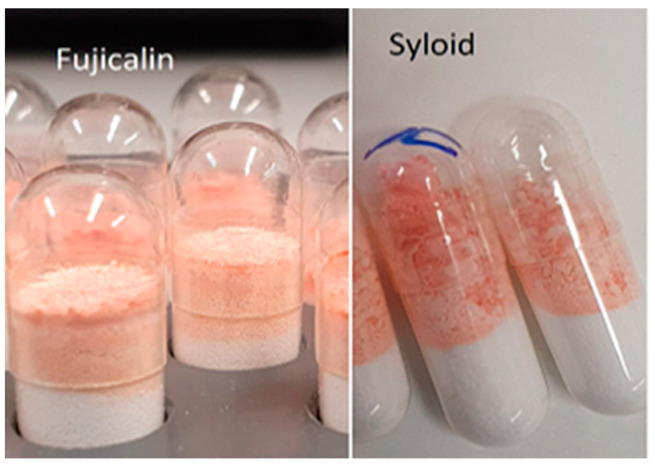
Inhomogeneous distribution of API solution (colored).

**Figure 5 pharmaceutics-14-01580-f005:**
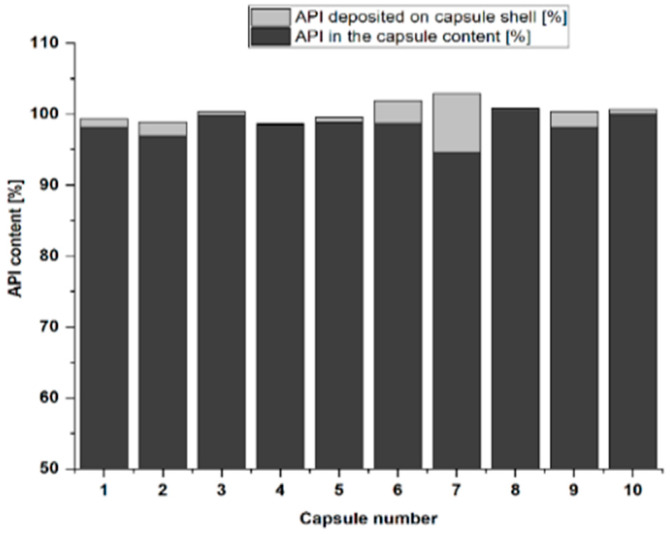
Total API content (%) in the capsules of 1 mg enalapril maleate with Syloid 244 FP, differentiated in the API deposited on the capsule shell and API in the capsule content (*n* = 10, dosing device: Hamilton syringe).

**Figure 6 pharmaceutics-14-01580-f006:**
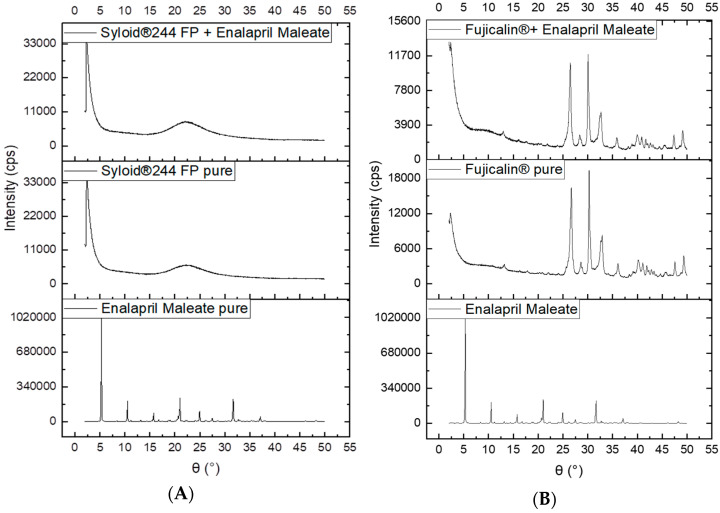
X-ray diffraction patterns of 1 mg enalapril maleate capsules prepared with Syloid 244 FP (**A**) and Fujicalin (**B**).

**Figure 7 pharmaceutics-14-01580-f007:**
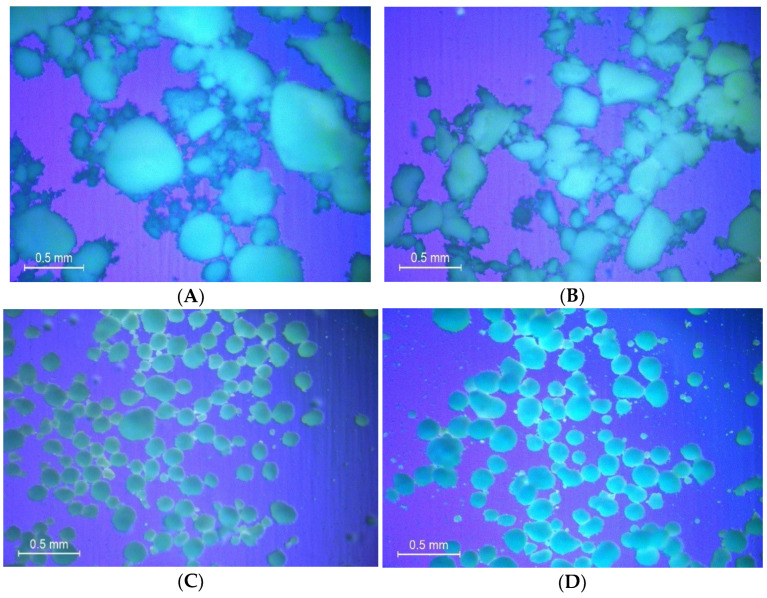
Images from polarized light microscopy on powder morphology of API and excipients used in capsule preparation. (**A**) Pure Syloid 244 FP. (**B**) Syloid 244 FP + enalapril maleate. (**C**) Pure Fujicalin. (**D**) Fujicalin + enalapril maleate. (**E**) Pure enalapril maleate.

**Table 1 pharmaceutics-14-01580-t001:** Enalapril maleate capsule formulation.

Item	Amount (per Capsule)
Hard gelatin capsules (Coni-Snap, size 0)	-
Enalapril maleate	1 mg
Amorphous silicon dioxide (Syloid 244 FP)	Approx. 0.68 mL
Dibasic calcium phosphate anhydrous (Fujicalin)	Approx. 0.68 mL
Ethanol (absolute)	Until required amount

## Data Availability

All data presented in this study are available at the Institute of Pharmaceutics and Biopharmaceutics at Heinrich Heine University Düsseldorf.

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
