# Peer review of "Implementation of a New Electronic Liquid Dispensing System for Individualized Compounding of Hard Capsules"

_pharmaceutics, 2022, doi:10.3390/pharmaceutics14081580_

Round 1

Reviewer 1 Report

The paper is well written. One very important remark is the very limited number of references. It is adviced to extend this part of the paper.

Author Response

Dear reviewer, we would like to thank you for your feedback. We understand the criticism on the introduction and have included more sources on the background and to consider within this topic without losing the focus on the electronic device for capsule filling. .

Reviewer 2 Report

The work presented is an attempt to develop an on-demand dispensing system that may find use in the hospital or community pharmacy, akin to personalized medicine. Whilst the premise for this type of investigation is warranted, the work is devoid of sufficient scientific data/experimentation.

Authors state that the research qualifies as GMP but did not give the evidence for this within the methodology.

The device used in the experiment was provided by GmbH and not fabricated by the authors so scientific ownership/intuitiveness of the research is weak.

For the creating of the matrix, the procedure described could have been well served using placebo tablets that are absorptive enough to allow the drug solution to be adsorbed on. In terms of speed, this will be more practical to the demands in the hospital of community pharmacy settings. Furthermore, even though alcohol is used to solubilise the API, it takes 3hrs to dry it off after instilling in the adsorbent within the capsule. Again, the capsule restricts free evaporation of solvent.

Authors contend that not all the alcohol is removed. This could lead to undesirable effects, therapeutically.

Adsorption of API is only in the upper part of the adsorbent material. This could be consequential with regard to API release in dissolution studies.

For study of this kind, drug release studies is crucial, yet this was not conduced here.

One of the concluding remarks by authors is that there were no API-excipient interaction observed however no study specific to ascertaining drug excipient interaction (such as FTIR or DSC analysis) was conducted.

Author Response

Dear Reviewer, thank you  very much for the feedback in order to improve our manuscript. We would like to answer in detail as follows.

The fact that the device can be used within GMP environment is a misconception. We have not tried or tested this, but have followed FDA guidelines to verify the device, which could enable for possible GMP compliance in the future. Of course, further testing would need to be done to verify this. We can understand that our text might be misleading, why we corrected the statement.

This work has been a collaboration with the company Waldeck & Saar. We were provided with the prototype of the machine, which is not yet commercially available on the market. We examined the machine in the context of this work and gave feedback on which adjusting screws in the system need to be improved, which the company then introduced into the system. We have not explained this sufficiently in the article and have supplemented it now.

Your suggestion to use placebo tablets is an interesting idea. However, the absorption profile of the tablet could be a limiting factor here, as it offers a rather small surface area.  The drying time would be even higher as a result. Added to this would be the time required to develop suitable tablets. Therefore, we decided to use a powder bed to maximize the absorption surface and to shorten the drying time. The fact that the capsule is not optimally suitable due to the different capsule walls is an important point that still requires improvement. 

We agree with your criticism of the unprecise drying time needed for the capsule content. At present, we are not able to shorten this time, which is why we still a need for improvement here.

The alcohol evaporation was checked gravimetrically . As ethanol is a solvent of class 3 Ph. Eur. (Low toxicity), loss on drying by mass was checked on your advice. The findings are also added into the corrected version. The qualification of ethanol should be performed using head-space gas chromatography, unfortunately this is currently not available to us. From the study, it can be concluded that ethanol concentration should be low as much as possible in the capsules made with Syloid®244 and formulation with Fujicalin® would be recommended.

We agree with you that release is an important parameter, especially as we work with porous excipients. To ensure appropriate release of the drug substance, we have performed XRPD analysis and polarized light microscopy analysis of capsule contents and a preliminary dissolution study was also performed to confirm the findings from XRPD and polarized microscopy analysis on your advice. The results can be found in the corrected version.

We have done HPLC analyses to get an idea of whether there are any interactions between the active ingredients. Such interaction products or degradation products could be analyzed by peak shifts or the appearance of further peaks. In none of the chromatograms typical peaks could be found, except the peak of the drug substance. FTIR would certainly be a good solution to confirm this again, unfortunately this is not available to us.

We hope that we could answer your questions sufficiently and that the corrected version will give you all the necessary information.

Reviewer 3 Report

This is a potentially good concept as the treatment of patients are being personalized. The manuscript is well written and the experiments reported are appropriate. I have some concerns about the determination of the quality of the capsules. Content uniformity although important is not the only concern.  What about:

  1. The stability of the prepared stock solutions of the drug
  2. The release profile of the drug from the capsules compared to commercially available products

Author Response

Dear Reviewer, Thank you very much for your sound feedback. The stock solution must always be prepared freshly to ensure suitable stability. Otherwise, preservatives must be added, which we wanted to avoid because of the unwanted potentially adverse effects, especially for paediatric patients.

As far as the release profiles are concerned, we fully agree with you. This aspect is very important although the drug under study is freely soluble in the specific case and was not affected at all by the excipients in preliminary investigations. To investigate the solid state of the drug substance, we have performed XRPD analysis and polarized light microscopy analysis of capsule contents  was also performed to confirm the findings from XRPD and polarized microscopy analysis. We have discussed the results from our investigation in the corrected version and added a precautionary statement on possible drug-excipient interactions affecting the drug release. We hope that we have been able to sufficiently answer your questions and promise to work on more critically drug substances in the concept of the novel device.

Round 2

Reviewer 2 Report

All comments posed have been addressed.

Author Response

Thank you very much for your positive feedback and the efforts in reviewing our paper.

Reviewer 3 Report

The article, although addressing a nitch area within pharmacy practice, represents a new approach to preparing individualized capsule dosage forms. The revised paper addresses all the reviewer's concerns.

Author Response

(The authors gave the same response as above.)
